# Optimizing video data security: A hybrid MAES-ECC encryption technique for efficient internet transmission

**Sobia Shafiq**[1,2], **Sohaib Latif**[3]*, **Jawad Ibrahim**[2], **M. Saad Bin Ilyas**[3], **Azhar Imran**[4], **Natalia Kryvinska**[5], **Ahmad Alshammari**[6]*, **Mohammed El-Meligy**[7,8]

**1** Faculty of Information Systems and Applied Computer Sciences, Otto-Friedrich-University Bamberg, Bamberg, Germany, **2** Department of Computer Science, National University of Modern Languages (NUML), Rawalpindi, Pakistan, **3** Department of Computer Science, The University of Chenab, Gujrat, Pakistan, **4** Department of Creative Technologies, Air University Islamabad, Islamabad, Pakistan, **5** Faculty of Management, Comenius University, Bratislava, Slovakia, **6** Department of Computer Sciences, Faculty of Computing and Information Technology, Northern Border University, Rafha, Kingdom of Saudi Arabia, **7** Jadara University Research Center, Jadara University, Irbid, Jordan, **8** Applied Science Research Center, Applied Science Private University, Amman, Jordan

\* sohaib@cs.uchenab.edu.pk (SL); ahmad.almkhaidsh@nbu.edu.sa (AA)

**Data Availability Statement:** All relevant data are within the manuscript.

**Funding:** The authors extend their appreciation to the Deanship of Scientific Research at Northern

## Abstract

Data security is becoming important as the amount of video data transmitted over the internet grows rapidly. This research article aims to maximize the security of transmitted video data by proposing a novel hybrid technique for video encryption and decryption. Elliptic Curve Cryptography (ECC) and the Modified Advanced Encryption Standard (MAES) are two encryption techniques that are included in the hybrid approach. By providing a more effective and safe method for video encryption and decryption, this research considerably advances the field of video data protection in Internet communication. In the proposed technique the video frames are extracted, and each frame is first encrypted using MAES technique and then again encrypted using ECC technique. After the encryption, the individual frames are merged to make an encrypted video. The same process is performed in reverse order to perform decryption of the video. The results of the experiments demonstrate the effectiveness of the suggested scheme: higher security, better accuracy, and shorter processing times when compared to well-known techniques such as Advanced Encryption Standard (AES), MAES, ECC, Simplified Data Encryption Standard (SDES), and Chaotic Map methods.

## Section 1: Introduction

With the expansion of internet and computer networks, video sharing in multimedia applications has now become an indispensable part of our lives. Now video data is being transferred in every field, including commercial, industrial, military, medical, etc. [1]. The security of such data has become a big concern as the data might contain some sensitive information. There is

Border University, Arar, KSA for funding this research work through the project number "NBU-FFR-2024-2990-01".

**Competing interests:** NO authors have competing interests.

a high need to secure sensitive information from intruders [2]. Cryptography aims to guard sensitive data from unauthorized users by converting it into an unrecognizable form [3].

In the past few years, researchers have proposed many video encryption techniques, including symmetric and asymmetric techniques. Symmetric technique makes use of similar key for encryption and decryption of data, while asymmetric technique uses different keys, one is for encryption and other is for decryption of the data. These techniques are different in terms of the security they provide, complexity in terms of computational, and the time they need for encryption and decryption of the video data [4]. With improvements to its fundamental design and key scheduling algorithm, the widely used AES has evolved into the MAES. MAES is a symmetric technique that provides high security for the encryption of data in large amount [5], The purpose of this change is to strengthen encryption and make it more resistant to new types of cryptographic assaults. However, ECC is an asymmetric technique with a short key that requires less storage space and is considered suitable for encrypting small amounts of data [6]. ECC is well known for being resource-constrained, especially in environments like multimedia applications, because of its efficiency in terms of both key size and computational complexity [7].

To strengthen video content security via strong encryption and effective decryption procedures, this research paper presents a novel hybrid approach that combines two potent cryptographic techniques: ECC and the MAES. By combining the advantages of both algorithms, MAES and ECC offer a comprehensive and sophisticated solution to the problems presented by modern video encryption requirements. Our hybrid approach aims to create a strong and secure framework for video encryption and decryption by combining the advantages of MAES and ECC. This study explores the technical details of MAES and ECC, examining their unique advantages and disadvantages before suggesting an integrated system that best utilizes their complementary qualities. The research holds importance as it can aid in the creation of sophisticated cryptographic methods designed for video content, tackling the difficulties brought about by the dynamic and substantial amounts of multimedia data. This paper's subsequent sections will examine the specific methodology, experimental findings, and comparative analyses, providing insight into the efficacy and possibility of the suggested hybrid approach in strengthening the security of video storage and communication in the digital sphere.

By combining the advantages of both techniques, we propose a hybrid technique for video encryption in which the videos are first encrypted using MAES and then the encrypted video is again encrypted using the ECC technique. Once the video is encrypted, the decryption of the video can be performed by applying both techniques to the encrypted video but in reverse order. Experimental results illustrate that the proposed hybrid technique is highly secure and takes less computational time than AES, MAES, ECC, SDES and Chaotic Map techniques.

The remaining paper is structured as follows. Section 2 comprises the literature review. Section 3 considers the proposed methodology in detail. Section 4 represents analysis of experimental results, and Section 5 provides the conclusion.

## Section 2: Literature survey

The proliferation of digital video content and the increasing reliance on internet transmission underscores the critical importance of video data security. As video data traverse the internet, they face numerous challenges, including unauthorized access, data interception, and potential compromise. Therefore, the optimization of video data security is imperative to ensure the confidentiality, integrity, and authenticity of transmitted content.

In this paper [8] author proposes a new hybrid multi-key cryptography technique for secure communication of video. The objective of the research is to address the concern of protecting

copyright and preventing piracy in real-time video streaming systems. The proposed technique utilizes the ECC method as a pseudorandom encryption key generator to encrypt and decrypt small chunks of video files dynamically. Multiple keys are generated on video-based data, enhancing the security of the encryption process. The implementation of the proposed technique was carried out on the Android platform, with sender and recipient applications developed for streaming videos. The performance and security of the system were evaluated, and the outcomes demonstrated superiority in terms of both aspects. The results show significant improvements in terms of encryption and decryption time, as well as parameters such as SSIM, SNR, PSNR, MSE, and RMSE. In [9] author proposes a secure video communication technique based on multi-equation multi-key hybrid cryptography that helps to improve the security of video communication. The proposed technique aims to secure the content of video from unauthorized access and ensure the confidentiality and integrity of the transmitted data. The results demonstrate the effectiveness of the multi-equation multi-key hybrid cryptography approach in providing enhanced security for video communication. This research [10] presents a hybrid cryptographic approach that combines the AES and DES algorithms. The paper discusses the implementation of the hybrid cryptographic technique and evaluates its performance and security. The results exhibit the effectiveness of the hybrid approach in terms of encryption and decryption speed and the level of security provided.

A video encryption scheme that utilizes hybrid encryption technology is presented in this research [11]. The focus of this research is to enhance the security of video content by employing a combination of encryption techniques. The proposed scheme aims to protect video data from unauthorized access and ensure the confidentiality and integrity of the content. The results demonstrate the effectiveness of the hybrid encryption technology in providing robust protection for video content. The objective of the research in [12] is to obtain an encryption algorithm which combines the efficiency of MAES and the security of ECC. The paper describes the implementation of the EMAES algorithm in MATLAB and Android Studio, using a messaging application. The results show that EMAES is 30% more efficient in terms of encryption and decryption time compared to other algorithms. The security of EMAES is also improved when compared to other hybrid algorithms, as demonstrated by parameters like, SNR, PSNR, SSIM, MSE, and RMSE.

The author [13] proposed a real-time video security system that utilizes a chaos-improved Advanced Encryption Standard (IAES) algorithm. This research enhances the security of real-time video streaming systems by incorporating chaos-based encryption techniques into the AES algorithm. The paper describes the implementation of the IAES algorithm and its application in video encryption and decryption. The IAES algorithm utilizes chaos-based techniques to generate encryption keys, which are used to encrypt and decrypt video data in real-time. The research evaluates the performance and security of the proposed system by implementing it on devices and streaming videos. In this paper [14] author proposes a multi-level image security system that combines ECC, magic matrix, and the AES. The objective of the research is to enhance the security of image data by utilizing multiple encryption techniques.

The proposed system utilizes ECC to generate encryption keys and achieve encryption and decryption operations on the image data. Additionally, a magic matrix is employed to further improve the security of the encrypted image. The AES algorithm is used as the primary encryption algorithm in the multi-level security system. The research evaluates the performance and security of the proposed system by conducting experiments on image data.

A novel approach for multimedia encryption using a hybrid cryptographic technique is proposed in [15]. The proposed approach utilizes a hybrid cryptographic technique, which combines multiple encryption algorithms to encrypt multimedia data. The research evaluates the performance and security of the proposed approach by conducting experiments on

multimedia data. The paper titled "A novel hybrid cryptosystem for secure streaming of high efficiency H.265 compressed videos in IoT multimedia applications" presents innovative hybrid cryptosystem for streaming of H.265 compressed videos securely in IoT multimedia applications. The objective of the research is to develop a secure cryptosystem that can be used for streaming high efficiency H.265 compressed videos in IoT multimedia applications. The proposed hybrid cryptosystem combines multiple encryption techniques to ensure the security of the streamed videos. The research evaluates the execution and security of the proposed cryptosystem by conducting experiments on H.265 compressed videos [16].

Author in [17] used the Advance Encryption Standard (AES) algorithm for video encryption. In their approach different sets of round keys were derived from the cipher keys and the plain text was initialized in a state array. The initial round key was then added to the start of the state array and ten different rounds were performed for the state modification. After these rounds, the final state array was copied as cipher text output. Different video lengths ranging from 18 to 52 seconds were used for experiments. In this paper [18] researchers highlighted the limitations of the AES technique for video encryption including high calculations, computation overhead and time consumption. They proposed an AES technique with modifications and named it as Modified Advance Encryption Standard (MAES). Experiments in [18] were performed using the same video lengths that were used in [17] and the comparison showed that the MAES technique worked faster as compared to the AES technique.

Shrutika et al. [19] presented a scheme which consisted of three modules. These modules are data embedding, video encryption, and data extraction. The video encryption was performed using the RC4 method, which is a stream cipher and produces improve results as compared to a block cipher. The RC4 algorithm was provided with a secret key to generate a key stream. XOR operation was then performed on the generated key stream and the video stream to perform video encryption. Although, the RC4 algorithm is a block cipher but is more vulnerable to the attacks and this technique cannot be easily implemented on small streams of data. In [20] author proposed a method of video encryption using the SDES and chaotic map schemes. In the proposed scheme, video was converted from RGB to Y-Cb-Cr color space to make encryption simpler. After the conversion, only the Y component was encrypted through a selection mechanism using the SDES, and the encryption on remaining frames was performed using a chaotic map scheme. For exchanging the keys, the RSA public key cryptosystem was used. At last, the merging of Y-Cb-Cr frames was done to make a video. The resultant video was then changed back to RGB to get an encrypted RGB video. The proposed technique takes much time for conversion from R-G-B to Y-Cb-Cr color space of videos which makes it a time-consuming technique.

In [21], researchers proposed a method called Unequal Secure Encryption (USE), which encrypts different portions of videos using different cryptographic techniques. The proposed scheme was divided into two modules. The First module was based on the classification of the video data, and the second module was based on USE. Two segments were generated in the video classification module which they named as "important" and "unimportant" video segments. After the division, the important video segments were encrypted using the AES technique and the unimportant video segments were encrypted using the Fast Leak Extraction (FLEX). After encrypting with FLEX, they were subjected to an XOR operation to reduce computational costs. Although the scheme offered advantages like low computational cost, but it is a quite complex technique. Authors used ECC technique for real time video encryption in [22]. The suggested approach focuses on analyzing the performance of several ECC curves for real-time video encryption before recommending an appropriate ECC curve for the best outcomes. The suggested method is based on a client-server architecture in which the server encrypts the video frame by frame as the client sends the initial request for video. It was

determined that the X9.62 272-bit binary curve and the NIST-recommended 256-bit prime curves are appropriate for real-time video encryption after looking at 18 different curves. The suggested method was developed only on an institutional network and is only mentioned for real-time video streaming. The literature review is discussed more in Table 1 with their contribution.

## Section 3: Proposed methodology

Video encryption technique must be secure, fast and easy to implement so that it can be used in real world. After reviewing different video encryption techniques, our proposed methodology is a hybrid video encryption and decryption technique in which the advantages of Modified Advance Encryption Standard (MAES) and ECC algorithms are used. In the proposed technique the video is split into multiple frames for encryption and decryption and an array of bitmap images is created which holds the frames of the video. After that each video frame is first encrypted using MAES algorithm and then again encrypted using ECC algorithm. The flowchart of proposed technique is given below shown in Fig 1.

MAES algorithm uses the concept of row shifting and ECC follows column shifting of pixels present in a frame. Shifting of pixels will depend on the grayscale value of each $4^{th}$ pixel in both vertical and horizontal direction which will be calculated using the equation 1.

$$G = \frac{r + g + b}{3} \tag{1}$$

Once encryption is done, the encrypted frames will be combined to generate a complete encrypted video. On the other hand, while performing decryption the encrypted video will be split into multiple frames and each frame will be decrypted using ECC and MAES algorithm and then combined into an understandable digital video. This section further provides an overview of encryption, decryption and key generation steps used in the proposed technique.

### 3.1. Encryption process in proposed methodology

Fig 2 represents the encryption process of the proposed technique where random keys are generated including MAES shared key, ECC public and ECC private key. These keys are used for the encryption and decryption of the video frames and to enhance the security to the video data, the MAES shared key and ECC private key are encrypted using Rail Fence Cipher [15]. The pseudocode of the proposed technique is described below:

To perform encryption, first the video is split into multiple frames and then each frame is first encrypted by MAES algorithm using its key. Once frame get encrypted using MAES algorithm, again encryption is performed on it using the public key of ECC algorithm to further encrypt the video frames. As each frame gets encrypted by MAES and ECC algorithms, it will be stored in a bit map array. When all the frames of video get encrypted, the bitmap array will be converted into encrypted version of the original video.

### 3.2. Decryption process in proposed technique

Fig 3 shows the decryption process of the encrypted video where the key file and encrypted video is used. To perform the decryption of video, the keys are first decrypted. The encrypted video is again split into frames and each frame is first decrypted using ECC algorithm and the decrypted frame is then again decrypted using MAES algorithm to get the original video.

**Table 1. Background study.**

| Ref. | Year | Method Used | Contribution | Limitation |
|---|---|---|---|---|
| [8] | 2023 | Hybrid Multikey Cryptography | Introduces an advanced hybrid multikey cryptography technique for securing video communication. | Specifics of the hybrid approach and its adaptability to various video formats may require further exploration. |
| [9] | 2023 | Multi-Equation Multi-Key Hybrid Cryptography | Presents a secure video communication system utilizing a multi-equation multi-key hybrid cryptography approach. | The effectiveness in diverse network conditions and scalability could be areas for further investigation. |
| [10] | 2023 | Hybrid Cryptographic Encryption and Decryption | Proposes a hybridized encryption and decryption technique using both AES and DES for enhanced security. | The performance trade-offs and computational efficiency of the hybrid scheme might need further analysis. |
| [11] | 2020 | Hybrid Encryption Technology | Introduces a video encryption scheme utilizing hybrid encryption technology for enhanced data security. | Specifics on the adaptability to various video formats and real-time performance could be areas for exploration. |
| [12] | 2023 | EMAES Hybrid Encryption Algorithm | Implements and evaluates the EMAES hybrid encryption algorithm for secure and efficient multimedia file sharing. | Further investigation is required to assess the algorithm's performance in large-scale multimedia file sharing. |
| [13] | 2022 | Chaos-Improved Advanced Encryption Standard (IAES) | Proposes a real-time video security system employing IAES, enhancing the security of video streams with chaos. | The impact on computational overhead and real-time performance in different scenarios may require examination. |
| [14] | 2022 | Elliptic Curve, Magic Matrix, AES | Introduces a multi-level image security system combining elliptic curve, magic matrix, and AES for heightened security. | The scalability and efficiency of the proposed multi-level security system may warrant further exploration. |
| [15] | 2016 | Hybrid Cryptographic Techniques | Presents a novel idea of multimedia encryption using hybrid cryptographic techniques, offering enhanced security. | Specifics on the adaptability to different multimedia formats and real-time performance could be areas for exploration. |
| [16] | 2020 | Hybrid Cryptosystem for Secure Streaming | Proposes a novel hybrid cryptosystem for secure streaming of compressed videos in IoT multimedia applications. | The impact on latency and resource utilization in IoT environments may need further investigation. |
| [17] | 2014 | AES Algorithm | Discusses video encryption using the AES, contributing to video data security. | The adaptability and efficiency of the AES algorithm in different video formats and sizes could be explored. |
| [18] | 2014 | Modified AES Algorithm | Introduces a modified AES algorithm for MPEG video encryption, aiming to enhance security in multimedia applications. | The applicability of the modified AES algorithm to various MPEG video formats and compression levels may need scrutiny. |
| [19] | 2015 | RC4 Encryption Scheme | Discusses securing compressed video streams using the RC4 encryption scheme, contributing to multimedia data security. | The robustness and performance of the RC4 encryption scheme in different compression scenarios could be explored. |
| [20] | 2018 | Selective Frame Scheme | Proposes robust encryption of uncompressed videos with a selective frame scheme, contributing to video data security. | The efficiency and trade-offs of the selective frame scheme in different video content and sizes may warrant investigation. |
| [21] | 2007 | New Video Encryption Scheme for H. 264/AVC | Introduces a new video encryption scheme specifically designed for H. 264/AVC, enhancing video data security. | The compatibility and performance of the proposed scheme with other video compression standards may need examination. |
| [22] | 2018 | Elliptic Curves for Real-time Video Encryption | Conducts a performance analysis of elliptic curves for real-time video encryption, contributing to video security. | The scalability and resource utilization of elliptic curves in real-time scenarios may need further exploration. |
| [23] | 2022 | Secure Image Steganography Technique based on Hybrid Transforms and Enhanced AES | Proposing an innovative approach to image steganography that utilizes hybrid transforms and enhanced Advanced Encryption Standard (AES) algorithms. | The paper aims to enhance the security and robustness of steganographic methods for hiding secret information within digital images. |
| [24] | 2019 | Chaos based efficient selective image encryption | Discusses the significance of encryption in ensuring patient privacy and the secure transmission of medical data. | It underscores the importance of encryption in maintaining patient privacy and ensuring the secure transmission of medical data. |
| [25] | 2018 | Elliptic Curve, Multi-Keys, Chaotic Map | Introduces elliptic curve video encryption based on multi-keys and chaotic map for secure mobile communication. | The scalability and computational efficiency of the proposed method in mobile environments may require further examination. |
| [26] | 2023 | Encryption algorithm, Chaos-based block permutation | The proposed encryption algorithm can effectively resist the chosen-plaintext attack. | It is crucial to recognize the potential limitations and continuously assess its security posture in evolving threat landscapes. |

*(Continued)*

**Table 1.** (Continued)

| Ref. | Year | Method Used | Contribution | Limitation |
|------|------|-------------|--------------|------------|
| [27] | 2020 | Cryptanalyzing an image cipher using multiple chaos and DNA operations | A chosen-plaintext attack method to attack ICIC-DNA. Differential analysis is firstly adopted to break the DNA-base permutation process, and then the DNA domain encryption is eliminated, and finally the equivalent key is used to achieve complete cracking. | The limitations stem from the lack of detail, context, and discussion on ethical and legal considerations, as well as ambiguity in terminology. These factors hinder a thorough understanding and evaluation of the text's claims and implications. |
| [28] | 2023 | Bit-level image encryption algorithm based on chaotic maps | Chooses a cipher image with the same sum value as that of the target cipher image to break the confusion module possessing a dynamic mechanism. | The absence of real-world application scenarios and comparative analyses diminishes the study's relevance and practical significance in cryptography. |
| [29] | 2022 | Cryptanalysis of an image encryption algorithm using quantum chaotic map and DNA coding | Revealing weaknesses in image encryption through cryptanalysis of an algorithm utilizing a quantum chaotic map and DNA coding. | Encryption algorithm using quantum chaotic map and DNA coding is inherently secure, overlooking vulnerabilities or weaknesses that may exist under certain conditions or attack scenarios. |
| [30] | 2024 | Image encryption scheme using variant Hill cipher and chaos | The research paper contributes by uncovering vulnerabilities through cryptanalysis of an image encryption scheme that utilizes a variant of the Hill cipher and chaos. | Encryption scheme using a variant of the Hill cipher and chaos is secure, without fully exploring potential vulnerabilities or weaknesses that may exist under different attack scenarios or cryptographic analyses. |

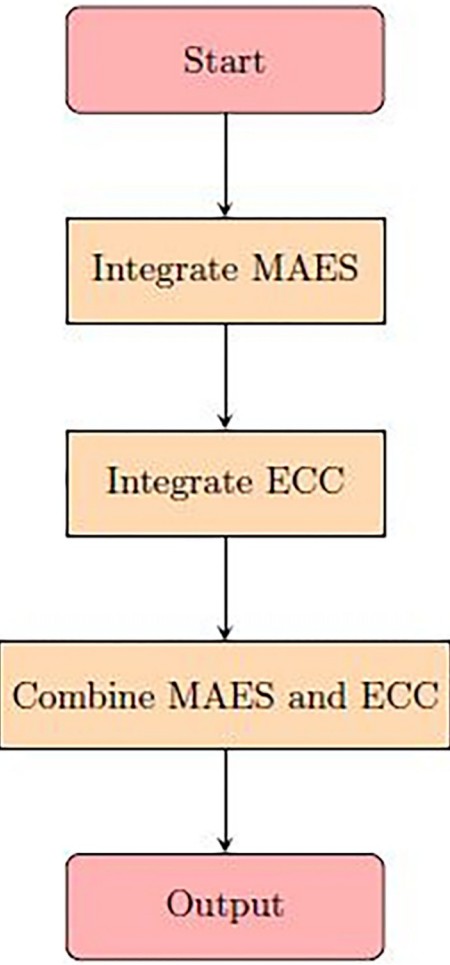

**Fig 1. Flowchart of proposed technique.**

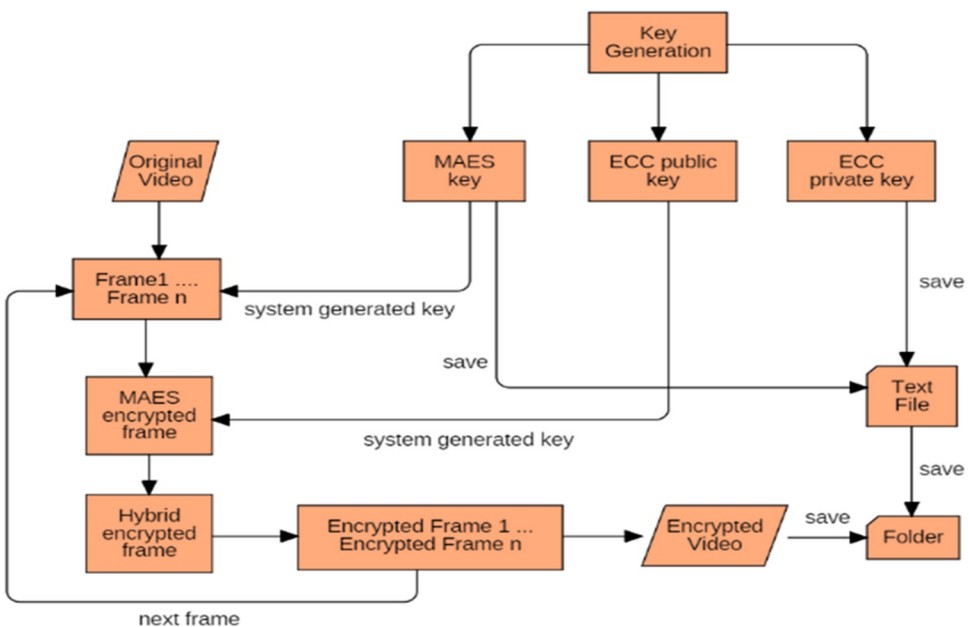

**Fig 2. Encryption process in proposed technique.**

| **Pseudocode for MAES Encryption** |
|---|
| function MAES_Encrypt(plaintext, key): |
| expandedKey = KeyExpansion(key) |
| state = AddRoundKey(plaintext, expandedKey[0]) |
| for round from 1 to Nr-1: |
| state = SubBytes(state) |
| state = ShiftRows(state) |
| state = MixColumns(state) |
| state = AddRoundKey(state, expandedKey[round]) |
| state = SubBytes(state) |
| state = ShiftRows(state) |
| ciphertext = AddRoundKey(state, expandedKey[Nr]) |
| return ciphertext |
| **Pseudocode for ECC Encryption** |
| function ECC_Encrypt(plaintext, publicKey): |
| messagePoint = EncodeMessageAsPoint(plaintext) |
| ephemeralKey = GenerateEphemeralKey() |
| sharedSecret = Multiply(publicKey, ephemeralKey) |
| ciphertext = PointAddition(messagePoint, sharedSecret) |
| return ciphertext |
| **Integration of MAES and ECC Encryption** |
| function Hybrid_Encrypt(plaintext, maesKey, eccPublicKey): |
| // Step 1: MAES Encryption |
| maesCiphertext = MAES_Encrypt(plaintext, maesKey) |
| // Step 2: ECC Encryption |
| eccCiphertext = ECC_Encrypt(maesCiphertext, eccPublicKey) |
| return eccCiphertext |

## 3.3. Operation on each individual frame

In the proposed technique MAES technique is used for row shifting of individual frames of video. In MAES, the value of gray value of $1^{st}$ row and $1^{st}$ column is checked. If the gray value of the $1^{st}$ row and the $1^{st}$ column is even, then no shift will be performed in the $1^{st}$ and $4^{th}$ row

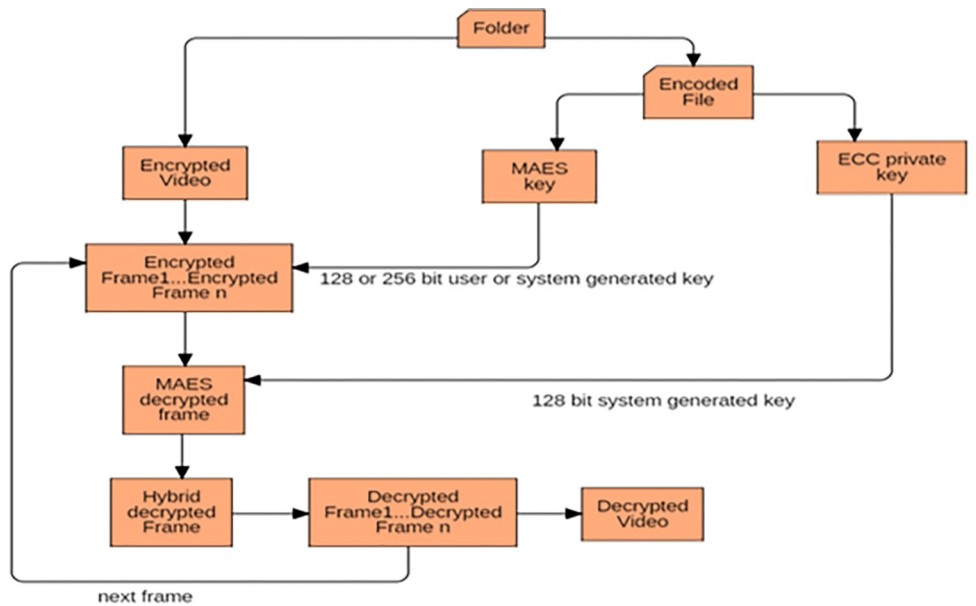

**Fig 3. Decryption process in proposed technique.**

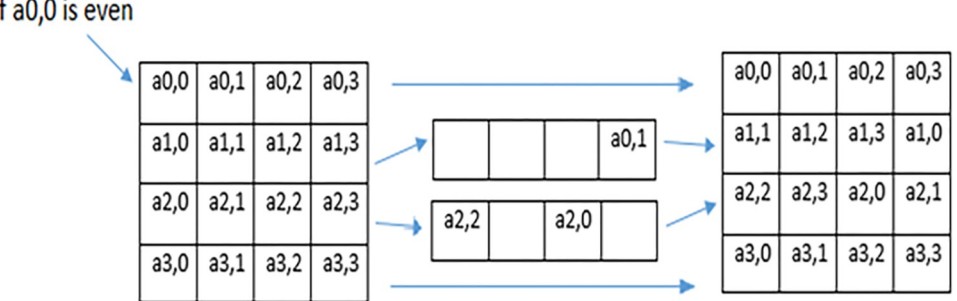

**Fig 4. MAES Row shifting when 1st row and 1st column is even.**

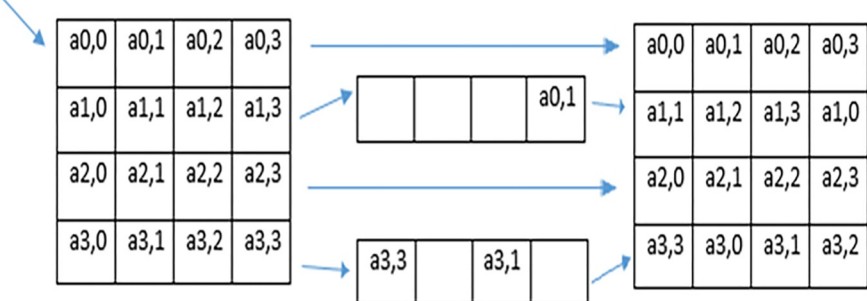

**Fig 5. MAES row shifting when 1st row and 1st column is odd.**

however 2$^{nd}$ row is shifted towards the right three places cyclically, and the 3$^{rd}$ row is shifted towards the left five places cyclically as shown in Fig 4.

On the other hand, if the value of 1$^{st}$ row and 1$^{st}$ column is odd, then the 1$^{st}$ and 3$^{rd}$ row remain unchanged but the 2$^{nd}$ row is shifted towards left by one place and the 4$^{th}$ row is shifted towards left three place as shown in Fig 5. The same operation will perform on each frame 14 times i.e. 14 rounds for MAES encryption.

Once the MAES encryption is performed, the MAES encrypted frame will be further encrypted using ECC's algorithm. In the proposed technique ECC algorithm is used to perform the column shifting of the pixels in the video frames which is made in pixels vertically upward or downward.

```
Algorithm 1:MAES and ECC encryption on each frame of a video
def encrypt_video(video_frames):
frameCount = len(video_frames)
e = keyGeneration(ECC_keys)
m = keyGeneration(MAES_key)
encrypted_frames = []
for i in range(frameCount):
frame = video_frames[i]
# MAES encryption
frame = shiftRow(frame, m)
frame1 = MAES_encryption(frame)
# ECC encryption
frame1 = shiftColumn(frame1, e)
frame2 = Hybrid_encryption(frame1)
encrypted_frames.append(frame2)
return encrypted_frames
```

## Section 4: Results

In the proposed methodology a hybrid technique is proposed which comprises of two different algorithms including MAES and ECC for encryption and decryption of a video. The proposed technique is developed using C sharp. Experimentation is performed on the proposed technique using different video sizes that are already used in the different research papers and the analysis of results is discussed in this section.

Table 2 represents the comparison of encryption time required for encrypting a video using AES algorithm and our proposed hybrid technique. Result revels that our system takes much less encryption time as compare to the existing technique [17].

Table 3 shows the comparison of the encryption and decryption time required for encrypting and decrypting a video using MAES [18] algorithm and proposed technique.

Table 4 represents the comparison of the encryption and decryption time required to encrypt and decrypt different video length using the Hybrid technique 2018 (SDES and

**Table 2. Comparison between AES [17] and proposed technique.**

| Video Length | Encryption Time Using | |
|---|---|---|
| | AES | Proposed Hybrid System |
| 32 sec | 270 sec | 64.99 sec |
| 52 sec | 318 sec | 98.62 sec |
| 33 sec | 270 sec | 60.44 sec |
| 23 sec | 215 sec | 40.75 sec |
| 20 sec | 201 sec | 42.18 sec |

Table 3. Comparison of encryption and decryption time using MAES [18] and proposed technique.

| Video Size | MAES Time (ms) | | Proposed Hybrid Scheme (ms) | |
|---|---|---|---|---|
| MB | Encryption | Decryption | Encryption | Decryption |
| 1.26 MB | 1122 | 2540 | 1045 | 2341 |
| 4.45 MB | 2476 | 4823 | 2275 | 4652 |
| 1.11 MB | 917 | 2011 | 857 | 1987 |

Table 4. Comparison between Hybrid technique 2018 [19] and our proposed system.

| Video Length | Hybrid Technique 2018 (SDES & Chaotic Map Scheme) | | Our Proposed Hybrid System | |
|---|---|---|---|---|
| | ET | DT | ET | DT |
| 300 frames | 249.47 sec | 191.73 sec | 173.88 sec | 128.31 sec |
| 400 frames | 286.77 sec | 274.14 sec | 103.69 sec | 99.37 sec |
| 240 frames | 445 sec | 317.95 sec | 121.37 sec | 113.31 sec |
| 500 frames | 890.5 sec | 603.81 sec | 280.56 sec | 232.18 sec |

Table 5. Comparison of hybrid technique [23] and our proposed technique.

| File Size | Encryption Time in Seconds | |
|---|---|---|
| | Hybrid System (AES and ECC) | Our Proposed System |
| 1.1 Mb | 0.79 | 0.67 |
| 1.20 Mb | 0.85 | 0.78 |
| 4.45 Mb | 1.24 | 1.03 |

Table 6. Comparison of hybrid technique [24] and our proposed technique.

| Size of File | Hybrid Scheme | Proposed Technique |
|---|---|---|
| 896 KB | 0.16 | 0.11 |
| 1.31 MB | 0.24 | 0.20 |
| 1.5 MB | 0.30 | 0.24 |
| 3.75 MB | 0.64 | 0.53 |

Chaotic Map Scheme) [19] and our proposed system. Result shows that our proposed hybrid approach takes less time for encryption and decryption of same video length.

Table 5 shows the comparison of hybrid technique (including AES and ECC algorithm) discussed in [23] and our proposed technique.

Another hybrid technique is proposed in [24] comprises of AES and ECC algorithm. Table 6 shows the comparison of time required to encrypt and decrypt a video using technique discuss in [24] and our proposed technique.

Fig 6 represents the graphical comparison of time required for encryption using AES and our proposed hybrid scheme.

Fig 7 represents the graphical comparison of the encryption and decryption time required for encrypting and decryption same video lengths using the Hybrid Scheme 2018 (SDES and Chaotic Map scheme) and our proposed technique.

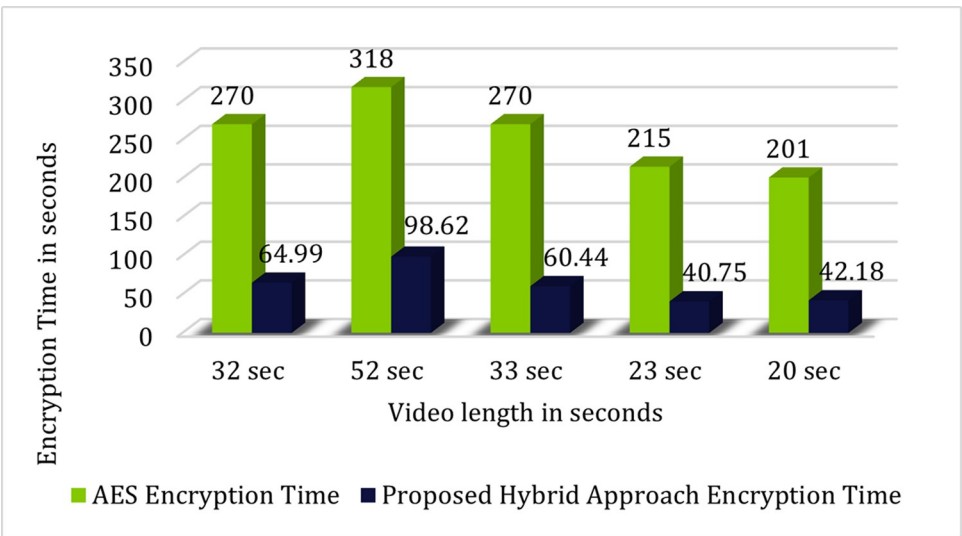

**Fig 6. Graphical comparison between AES and proposed hybrid system.**

## Section 5: Conclusion

This research paper has presented a novel approach for optimizing video data security through the development of a hybrid MAES-ECC encryption technique tailored for efficient internet transmission. By integrating Modified Advanced Encryption Standard (MAES) and Elliptic

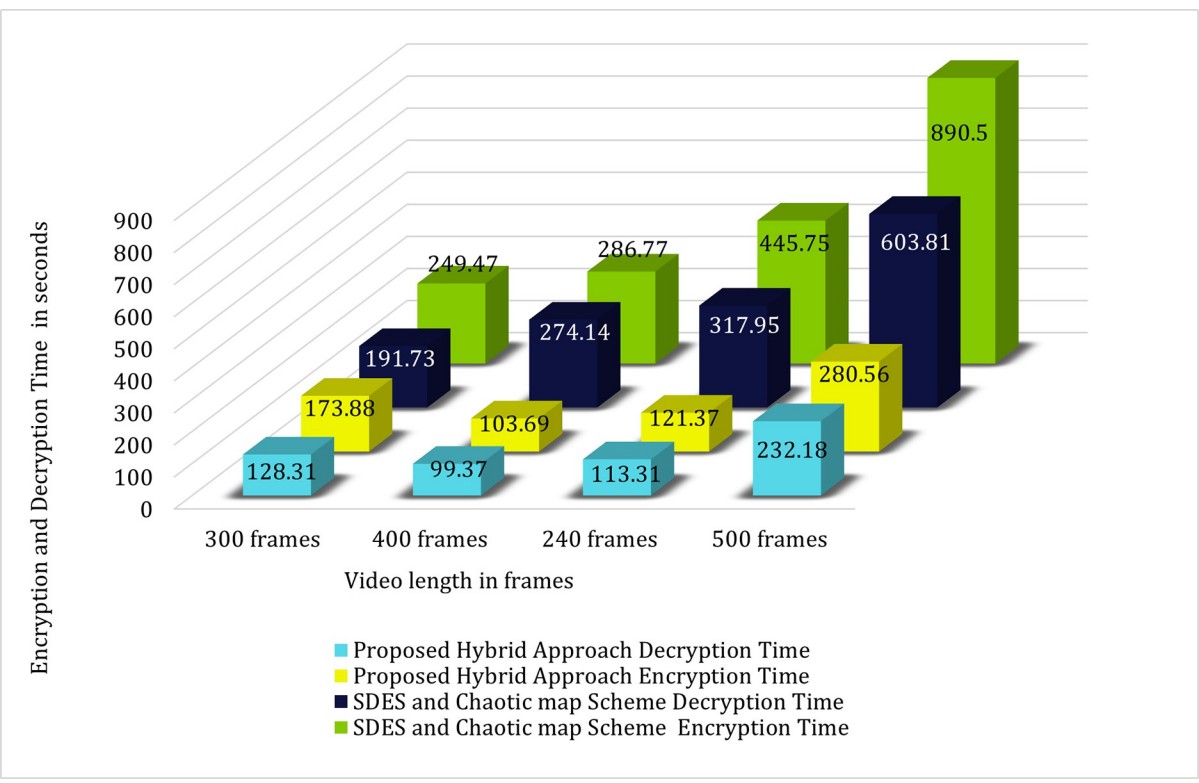

**Fig 7. Graphical comparison of the SDES and chaotic map scheme and the proposed hybrid system.**

Curve Cryptography (ECC), the proposed technique offers enhanced security measures while ensuring computational efficiency and facilitating seamless transmission over the internet. Throughout the study, we have elucidated the rationale behind key design decisions, including the selection of appropriate key sizes and encryption rounds, to address the unique challenges posed by video data encryption. Through extensive experimentation and analysis, the effectiveness and performance of the hybrid encryption technique have been demonstrated, underscoring its potential for safeguarding video data in diverse internet transmission scenarios. Moving forward, further research avenues may explore optimizations and enhancements to refine the proposed technique, thereby advancing the realm of video data security in the context of internet transmission.

## Acknowledgments

The first author is thankful to all co-authors who contributed to carrying out the proposed work.

## Author Contributions

**Conceptualization:** Sobia Shafiq, Jawad Ibrahim, Ahmad Alshammari.

**Data curation:** Sohaib Latif, Ahmad Alshammari.

**Formal analysis:** Sohaib Latif, Ahmad Alshammari.

**Funding acquisition:** Azhar Imran, Ahmad Alshammari.

**Investigation:** Jawad Ibrahim, M. Saad Bin Ilyas.

**Methodology:** Jawad Ibrahim, M. Saad Bin Ilyas, Mohammed El-Meligy.

**Project administration:** Azhar Imran, Mohammed El-Meligy.

**Resources:** Sohaib Latif, Natalia Kryvinska.

**Software:** Sobia Shafiq, Natalia Kryvinska, Mohammed El-Meligy.

**Supervision:** Azhar Imran, Natalia Kryvinska.

**Validation:** Mohammed El-Meligy.

**Visualization:** Sohaib Latif, M. Saad Bin Ilyas, Natalia Kryvinska.

**Writing – original draft:** Sobia Shafiq.

**Writing – review & editing:** Azhar Imran, Mohammed El-Meligy.

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
