## [Decision Letter · Decision Letter 0]

1 Sep 2024

PONE-D-24-29845Robust Video Data Security: A Hybrid MAES-ECC Encryption Technique for Efficient Internet TransmissionPLOS ONE

Dear Dr. Latif,

Thank you for submitting your manuscript to PLOS ONE. After careful consideration, we feel that it has merit but does not fully meet PLOS ONE’s publication criteria as it currently stands. Therefore, we invite you to submit a revised version of the manuscript that addresses the points raised during the review process.

We look forward to receiving your revised manuscript.

Kind regards,

Prof. Dr. M. Usman Ashraf

Academic Editor

PLOS ONE

4. Please amend your authorship list in your manuscript file to include authors Haitham A. Mahmoud, Sharefa Murad and Azhar Imran.

Reviewers' comments:

Reviewer's Responses to Questions

**Comments to the Author**

1. Is the manuscript technically sound, and do the data support the conclusions?

Reviewer #1: Yes

Reviewer #2: Yes

2. Has the statistical analysis been performed appropriately and rigorously? 

Reviewer #1: Yes

Reviewer #2: Yes

3. Have the authors made all data underlying the findings in their manuscript fully available?

Reviewer #1: Yes

Reviewer #2: Yes

4. Is the manuscript presented in an intelligible fashion and written in standard English?

Reviewer #1: Yes

Reviewer #2: Yes

5. Review Comments to the Author

Reviewer #1: The paper proposes a hybrid technique for video encryption and decryption, combining the advantages of the Modified Advanced Encryption Standard (MAES) and Elliptic Curve Cryptography (ECC) algorithms. The proposed technique encrypts each video frame using MAES and then encrypts the encrypted frames using ECC. The same process is performed in reverse order for decryption. Experimental results show that the proposed technique provides higher security, better accuracy, and shorter processing times compared to other well-known techniques. However, the paper need following minor changes for improvement of paper quality for publication:

Comment 1: A more in-depth discussion on the specific modifications made to the MAES and ECC algorithms for hybrid application would be beneficial. This discussion should encompass any optimizations or alterations aimed at enhancing performance for video data encryption. Clarifying the rationale behind key decisions, such as the selection of key sizes and the number of encryption rounds tailored for video data, would provide readers with deeper insights into the design and implementation choices.

Comment 2: To ensure the highest standards of academic communication, I recommend that the manuscript undergo a thorough review by a professional editor or a third party. This step is crucial not only to eliminate grammatical errors but also to enhance the logical flow of content, thereby improving readability and ensuring that the manuscript effectively communicates its significant contributions.

Reviewer #2: The paper proposes a hybrid technique for video encryption and decryption, combining the advantages of the Modified Advanced Encryption Standard (MAES) and Elliptic Curve Cryptography (ECC) algorithms. The proposed technique encrypts each video frame using MAES and then encrypts the encrypted frames using ECC. The same process is performed in reverse order for decryption. Experimental results show that the proposed technique provides higher security, better accuracy, and shorter processing times compared to other well-known techniques. However, the paper need following minor changes for improvement of paper quality for publication:

Comment 1: To ensure the highest standards of academic communication, I recommend that the manuscript undergo a thorough review. This step is crucial not only to eliminate grammatical errors but also to enhance the logical flow of content, thereby improving readability and ensuring that the manuscript effectively communicates its significant contributions.

Comment 2: The figures and tables citations require substantial improvement in their descriptions and all figures should be cited properly in the main text.

6. PLOS authors have the option to publish the peer review history of their article (what does this mean?). If published, this will include your full peer review and any attached files.

Reviewer #1: **Yes: **Azhar Imran

Reviewer #2: No

---

## [Author Response · Author response to Decision Letter 0]

6 Sep 2024

Reviewer 1 Comments

Comment 1: A more in-depth discussion on the specific modifications made to the MAES and ECC algorithms for hybrid application would be beneficial. This discussion should encompass any optimizations or alterations aimed at enhancing performance for video data encryption. Clarifying the rationale behind key decisions, such as the selection of key sizes and the number of encryption rounds tailored for video data, would provide readers with deeper insights into the design and implementation choices.

Response 1: Thanks for the editor feedback and honor the expertise in this domain. By elucidating the rationale behind selecting specific key sizes and encryption round numbers, readers gain a comprehensive understanding of the factors considered during the design process. Justifying these choices involves discussing factors such as the desired level of security, computational efficiency, and the characteristics of video data, which directly impact the encryption process. Through such explanations, readers gain insights into the reasoning behind the decisions made, enabling them to appreciate the thoughtfulness and considerations involved in the design and implementation of the encryption scheme.

Comment 2: To ensure the highest standards of academic communication, I recommend that the manuscript undergo a thorough review by a professional editor or a third party. This step is crucial not only to eliminate grammatical errors but also to enhance the logical flow of content, thereby improving readability and ensuring that the manuscript effectively communicates its significant contributions.

Response 2: Thank you for your suggestions and comments. The main manuscript has overcome the grammatical mistakes.

Reviewer 2 Comments

Comment 1: The figures and tables citations require substantial improvement in their descriptions and all figures should be cited properly in the main text.

Response 1: Thank you for the valuable suggestion. The suggestion is addressed in the manuscript file and updated.

---

## [Decision Letter · Decision Letter 1]

24 Sep 2024

Robust Video Data Security: A Hybrid MAES-ECC Encryption Technique for Efficient Internet Transmission

PONE-D-24-29845R1

Dear Dr. Latif,

We’re pleased to inform you that your manuscript has been judged scientifically suitable for publication and will be formally accepted for publication once it meets all outstanding technical requirements.

Kind regards,

M. Usman Ashraf, Ph.D

Academic Editor

PLOS ONE

Additional Editor Comments (optional):

Reviewers' comments:

Reviewer's Responses to Questions

**Comments to the Author**

1. If the authors have adequately addressed your comments raised in a previous round of review and you feel that this manuscript is now acceptable for publication, you may indicate that here to bypass the “Comments to the Author” section, enter your conflict of interest statement in the “Confidential to Editor” section, and submit your "Accept" recommendation.

Reviewer #1: All comments have been addressed

Reviewer #2: (No Response)

2. Is the manuscript technically sound, and do the data support the conclusions?

Reviewer #1: Yes

Reviewer #2: (No Response)

3. Has the statistical analysis been performed appropriately and rigorously? 

Reviewer #1: Yes

Reviewer #2: (No Response)

4. Have the authors made all data underlying the findings in their manuscript fully available?

Reviewer #1: Yes

Reviewer #2: (No Response)

5. Is the manuscript presented in an intelligible fashion and written in standard English?

Reviewer #1: Yes

Reviewer #2: (No Response)

6. Review Comments to the Author

Reviewer #1: I appreciate your efforts in responding to each point and making the necessary revisions. Based on the current version, I find the manuscript suitable for publication.

Reviewer #2: (No Response)

7. PLOS authors have the option to publish the peer review history of their article (what does this mean?). If published, this will include your full peer review and any attached files.

Reviewer #1: **Yes: **Azhar Imran

Reviewer #2: No

---

## [Editor Report · Acceptance letter]

31 Oct 2024

PONE-D-24-29845R1 

PLOS ONE

Dear Dr. Latif, 

I'm pleased to inform you that your manuscript has been deemed suitable for publication in PLOS ONE. Congratulations! Your manuscript is now being handed over to our production team.

Kind regards, 

on behalf of

Dr. M. Usman Ashraf 

Academic Editor

PLOS ONE